# Liquid crystal elastomer coatings with programmed response of surface profile

Greta Babakhanova [1,2], Taras Turiv [1,2], Yubing Guo [1,2], Matthew Hendrikx [3,4], Qi-Huo Wei[1,2], Albert P.H.J. Schenning [3,4], Dirk J. Broer [3,4] & Oleg D. Lavrentovich [1,2,5]

Stimuli-responsive liquid crystal elastomers with molecular orientation coupled to rubber-like elasticity show a great potential as elements in soft robotics, sensing, and transport systems. The orientational order defines their mechanical response to external stimuli, such as thermally activated muscle-like contraction. Here we demonstrate a dynamic thermal control of the surface topography of an elastomer prepared as a coating with a pattern of in-plane molecular orientation. The inscribed pattern determines whether the coating develops elevations, depressions, or in-plane deformations when the temperature changes. The deterministic dependence of the out-of-plane dynamic profile on the in-plane orientation is explained by activation forces. These forces are caused by stretching-contraction of the polymer networks and by spatially varying molecular orientation. The activation force concept brings the responsive liquid crystal elastomers into the domain of active matter. The demonstrated relationship can be used to design coatings with functionalities that mimic biological tissues such as skin.

[1] Liquid Crystal Institute, Kent State University, Kent, OH 44242, USA. [2] Chemical Physics Interdisciplinary Program, Kent State University, Kent, OH 44242, USA. [3] Functional Organic Materials and Devices, Department of Chemical Engineering and Chemistry, Eindhoven University of Technology, 5612 AZAE Eindhoven, The Netherlands. [4] Institute for Complex Molecular Systems, Eindhoven University of Technology, P.O. Box 513, 5600 MB Eindhoven, The Netherlands. [5] Department of Physics, Kent State University, Kent, OH 44242, USA. Correspondence and requests for materials should be addressed to O.D.L. (email: olavrent@kent.edu)

A liquid crystal elastomer (LCE) is an anisotropic rubber, as it is formed by cross-linked polymeric chains with rigid rod-like mesogenic segments in the main chain and attached as side branches; these mesogenic units are similar to the molecules forming low-molecular weight liquid crystals[1,2]. The mesogenic moieties in the nematic state of an LCE are oriented along a certain nonpolar direction called the director $\hat{\mathbf{n}} \equiv -\hat{\mathbf{n}}$. Cross-linked polymeric chains are structurally anisotropic because of their coupling to the orientational order. The coupling enables mechanical response of LCEs to external factors such as temperature. For example, upon heating, a uniformly aligned LCE strip contracts along the director and expands in the perpendicular directions, since the orientational order weakens and the cross-linked polymer network becomes more isotropic[1]. Such a uniform LCE strip behaves as an artificial muscle[3].

Recent research unraveled even more exciting effects when the director changes in space, $\hat{\mathbf{n}}(\mathbf{r}) \neq \text{const.}$ Thin LCE films with in-plane director patterns develop 3D shape changes with non-trivial mean and Gaussian curvatures when exposed to thermal or light activation[2,4–12], while director deformations across the film trigger wave-like shape changes and locomotion when activated by light illumination[13]. In a parallel vein, there is a tremendous progress in exploiting LCE coatings in which one surface is attached to the substrate and the other is free[14,15]. When illuminated with light, photoresponsive coatings with a misaligned director develop random spike-like topographies[16], while periodic elevations and groves can be produced by using cholesteric "fingerprint" textures[17], or periodic stripe arrays[18]. The challenge is in finding an approach by which the change of the topography of the coating or its stretching/contraction can be deterministically pre-programmed.

In this work, we demonstrate such programmed displacements and dynamics of surface profile in an artificial skin, made of an LCE. The program is written as a pattern of molecular orientation in the plane of the LCE coating that is initially flat. When activated by temperature, the coating changes its surface profile as prescribed by the in-plane pattern, by moving the material within

and out-of-plane. The displacements are deterministically related to the molecular orientation pattern. For example, circular bend of molecular orientation causes elevations, while radial splay causes depressions of the coating.

## Results

**Pre-programming the LCE coatings.** The director patterns with pre-designed splay and bend of the director are imposed onto the LCE by the plasmonic photoalignment technique[19] (see the section "Methods"). Two parallel glass plates separated by a distance of 5 μm coated with photosensitive azodye molecules are irradiated by a light beam that passes through a plasmonic photomask with a patterned array of elongated nanoapertures. The photomask imparts spatially-varied linear polarization onto the transmitted light beam[19]. Under irradiation, the azodye molecules reorient their long axes perpendicularly to the local light polarization[19]. When the liquid crystal in its monomeric state fills the gap between the two glass plates, the patterned surface orientation of the azodye molecules establishes the director pattern in the bulk. The mixture is then photopolymerized to obtain the LCE with the desired pattern of the director. The director is modulated in the $(x,y)$ plane of the film and remains parallel to the bounding plates; there is no change of the director through the thickness of the cell, as the top and bottom plate are photopatterned in the same way. One of the plates is removed to obtain the LCE coating. At this stage, the coating is flat.

Figures 1a, 2a, and 3a show the director patterns imprinted into the LCE coatings with arrays of topological defects of integer and semi-integer strength. The director field is of the form

$$\hat{\mathbf{n}} = (n_x, n_y, 0) = (\cos\alpha, \sin\alpha, 0) \qquad (1)$$

where $\alpha(x,y) = \sum_i m_i \arctan\left(\frac{y - y_{0i}}{x - x_{0i}}\right) + \varphi_0$, $m_i = \pm 1/2$, $\pm 1$ indicates the strength of the defect, $x_{0i}$ and $y_{0i}$ are the coordinates of the core of the defect and $\phi_0$ is the constant phase specifying the prevailing type of deformations and orientation of the defect

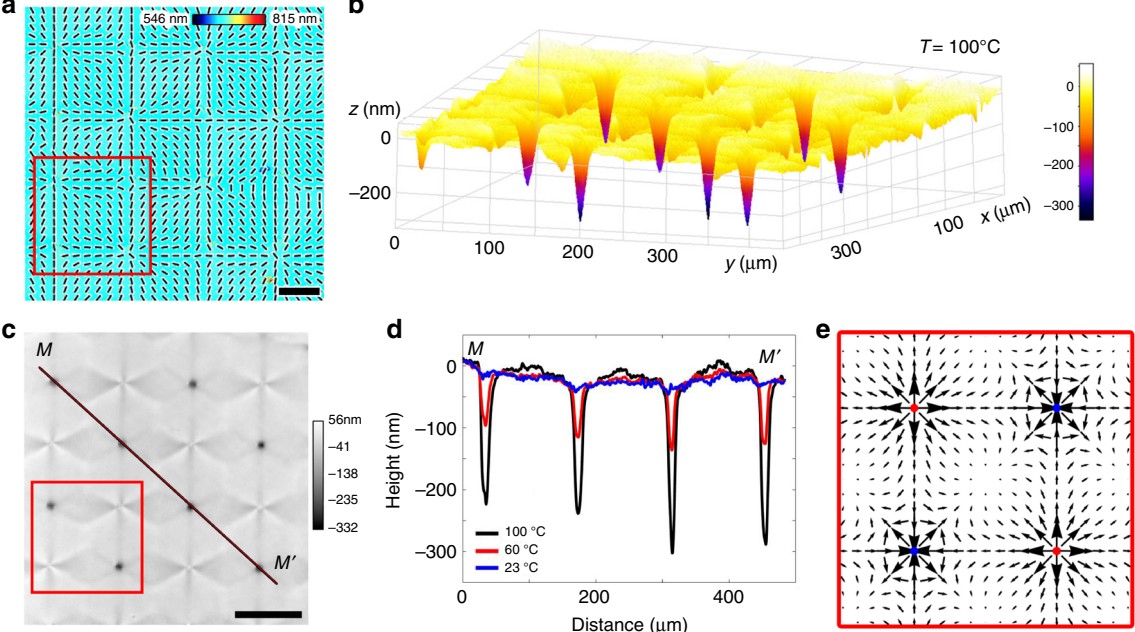

**Fig. 1** Surface depressions developed by radial defects upon heating. **a** PolScope image of a flat LCE coating at 23 °C which maps the optical retardance and the director orientation. Scale bar, 50 μm. **b** 3D image of surface topography of LCE coating at 100 °C with developed depressions at radial defects, as observed using DHM. **c** DHM image of the LCE surface at 100 °C used to extract the surface topography. Scale bar, 100 μm. **d** Surface profiles along line *MM'* in (**c**) at 23, 60, 100 °C. **e** Activation force density **f** map calculated for region outlined by a red box in (**a**, **c**)

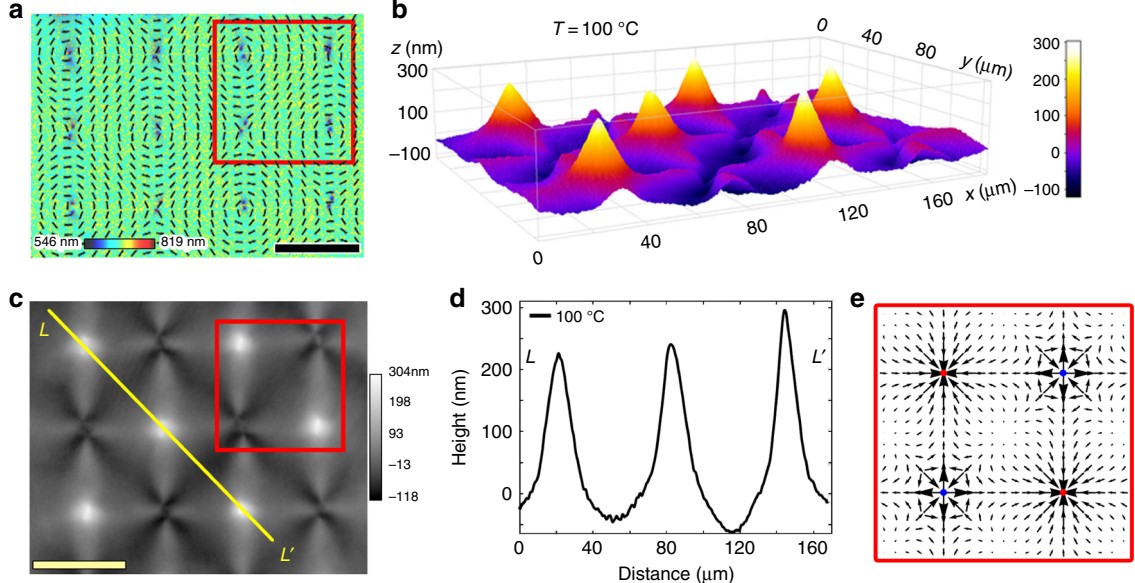

**Fig. 2** Surface elevations developed at circular defects upon heating. **a** PolScope image of a flat LCE coating at 30 °C which maps the optical retardance and the director orientation. Scale bar, 100 μm. **b** 3D image of surface topography of LCE at 100 °C with elevations developed at circular defects, as observed using DHM. **c** DHM image of the LCE surface at 100 °C used to extract the surface topography. Scale bar, 100 μm. **d** Surface profiles along line $LL'$ in (**c**) at 100 °C. **e** Activation force density **f** map calculated for region outlined by a red box in (**a**, **c**)

structures. The +1 defects are either of a radial type, carrying splay, when $\phi_0 = 0$ (Fig. 1a), or of a circular geometry, carrying bend, when $\phi_0 = \pi/2$ (Fig. 2a). The −1 defects in both Figs. 1a and 2a exhibit four alternating regions of splay and bend.

**Surface depressions caused by radial defects.** The LCE coating with the pre-inscribed director pattern is practically flat at the room temperature. The surface profile is established by digital holographic microscopy (DHM). When the coating is heated above the glass transition temperature (~50 °C), its surface topography shows a remarkably robust and reproducible change determined by the specific form of director distortions. Namely, splay regions associated with radial ($m = +1$ and $\phi_0 = 0$) defects in Fig. 1a, produce depressions (Fig. 1b–d). The material moves away from the center of the defect along the radial directions.

**Surface elevations caused by circular defects.** In contrast, bend regions associated with circular ($m = +1$ and $\phi_0 = \pi/2$) defects in Fig. 2a, produce pronounced hills (Fig. 2b–d). The amplitude of thickness variation is about 300–400 nm, representing a substantial fraction of the total thickness (5 μm). The typical aspect ratio for elevations in Fig. 2b,d, determined at 100 °C as the height of the deformation divided by its full width at half-height, was about 0.02; similar aspect ratio was found for depressions in Fig. 1b,d.

The heating-induced non-flat profile around $m = −1$ defects is more complicated, exhibiting four ridges and four valleys, which correlates with four regions of splay and four regions of bend (Figs. 1b, c and 2b, c). The dynamics of surface topography is completely reversible when the temperature is cycled between the room temperature and the maximum temperature of around 150 °C. The most pronounced profile deformations, such as depressions in Fig. 1b, d and elevations in Fig. 2b, d can lose their complete reversibility if the material is heated to temperatures above ~150 °C, presumably because of the developed irreversible strains, similarly to the case reported

by Ware et al.[7] for thermally addressed LCE films with two free surfaces.

**Coupled elevations-depressions caused by ±1/2 defects.** Similar relationships between the director patterns and surface topography are observed in patterns with $m = +1/2$ and $m = −1/2$ defects (Fig. 3). The defects $m = +1/2$ in Fig. 3 are of a polar symmetry with one bend region and one splay region; the $m = −1/2$ defects exhibit three regions of bend and three regions of splay each (Fig. 3a). The corresponding surface profile is comprised of a single elevation/depression pair in $m = +1/2$ case (Fig. 3b) and three elevations/depressions around the $m = −1/2$ defects, as expected from the symmetry of the defects. There is a new unique feature of the $m = +1/2$ defects, not observed neither for $m = −1/2$ nor for $m = ±1$ defects. Namely, upon heating, both the core and the elevation/depression pair associated with the $m = +1/2$ defect shift along the vector directed from the bend region towards the splay region (Fig. 3c, e, f). Figure 3e, f shows the temperature effect on the distance between $m = ±1/2$ defect pairs A and B, plotted as an increase (pair A) or a decrease (pair B) of their initial separation distance at 30 °C. The shift of $m = +1/2$ defect core is fully reversible in the heating-cooling cycles, provided the maximum temperature does not exceed ~150 °C.

The displacement of the elevation/depression pair is observed using DHM (Fig. 3c), by tracing the temperature dependent topography along $TT'$ line which crosses three defect cores: $−1/2, +1/2,$ and $−1/2$ (Fig. 3b). The inset in Fig. 3c demonstrates that on heating the LCE by 40 °C, the shift $\Delta y$ of elevation is about 5 μm. The displacement of the depression is about 3 μm. The associated shift of the $m = +1/2$ defect core is about 3 μm as observed under PolScope (Fig. 3e, f). The core is the central region of the defect at which the orientational order is strongly diminished. Its position is thus readily visualized by PolScope that maps the local optical retardance since the optical retardance at the core is much lower than in the rest of the sample (Fig. 3a, e)[20].

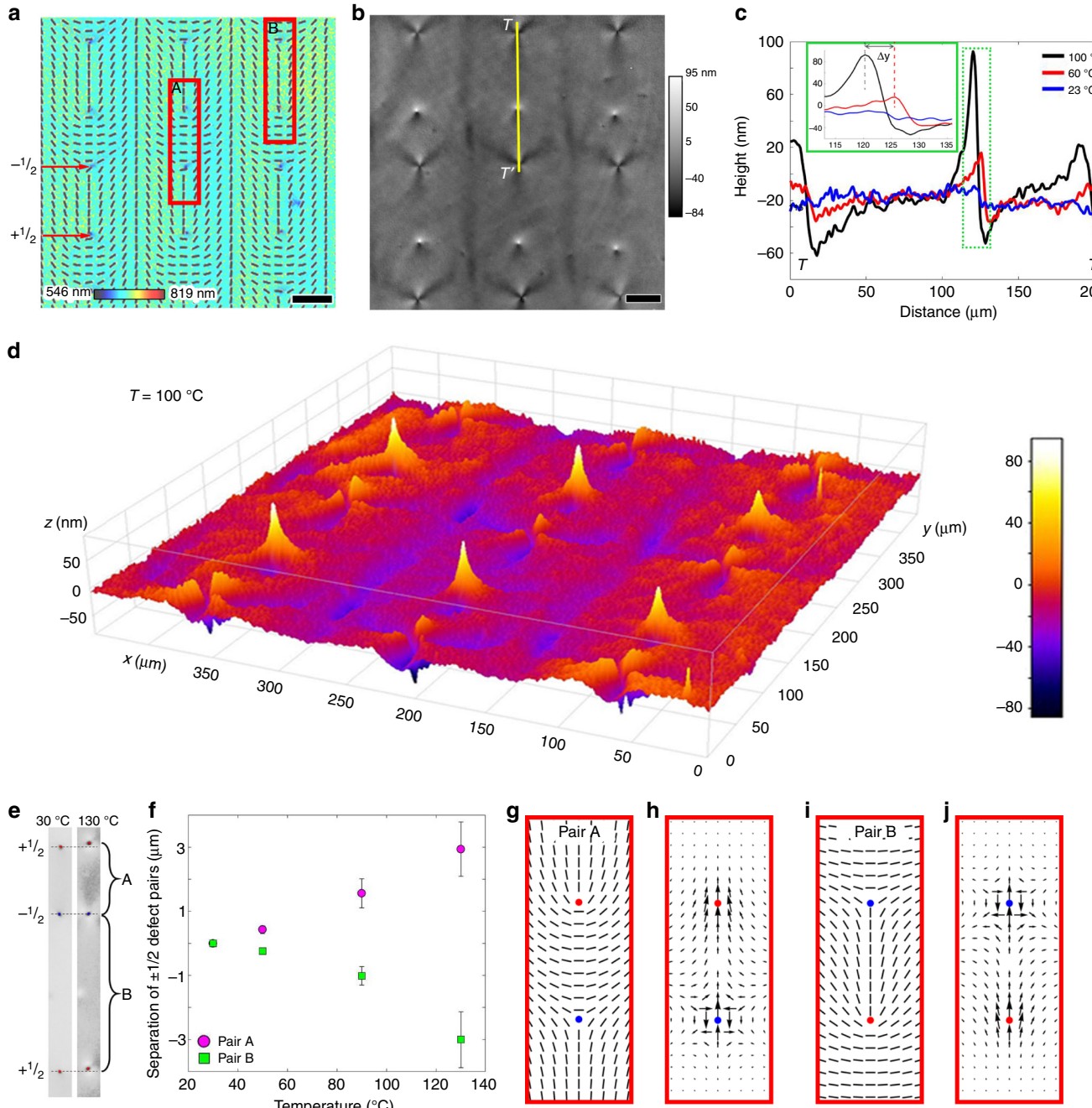

**Fig. 3** Coupled elevations–depressions caused by ±1/2 defects upon heating. **a** PolScope image of a flat LCE coating at 23 °C which maps the optical retardance and the director orientation; red boxes A and B show two different director configurations of the defect pairs. Scale bar, 50 µm. **b** DHM image of the non-flat LCE surface at 100 °C used to extract the surface profile along line *TT'*. Scale bar, 50 µm. **c** Surface profiles along line *TT'* in (**b**) at 23, 60, and 100 °C. **d** 3D image of surface topography of LCE at 100 °C observed using DHM. **e** Grayscale PolScope images taken at 30 and 130 °C showing the displacement of the half-integer defects as a function of temperature and director configuration that separates the defects. **f** Plot showing the distance separating ±1/2 defects as a function of temperature and director configurations A and B. The error bars represent the s.d.'s of 12 measurements. **g** Director configuration for pair A. **h** Activation force density **f** map calculated for the director pattern A. **i** Director configuration for pair B. **j** Activation force density **f** map calculated for director pattern B

**Depressions/elevations caused by splay/bend stripe patterns.** The correlations splay → depression and bend → elevation are observed not only in arrays with topological defects but also in defect-free patterns. As an example, Fig. 4 shows the response of an LCE coating with the splay-bend stripes, $\hat{n} = (n_x, n_y, n_z) = (|\cos\beta|, \sin\beta, 0)$, where $\beta(y) = \pi y/P$ and $P$ is half the period. The profile along the line *NN'* in the DHM texture (Fig. 4a) changes from being flat at room temperature to

strongly modulated at elevated temperatures, with alternating sharp valleys at the locations of a maximum splay and ridges at locations with a prevailing bend.

## Discussion

The experiments above demonstrate clearly that the heat-induced dynamic profile of an LCE coating is defined deterministically by

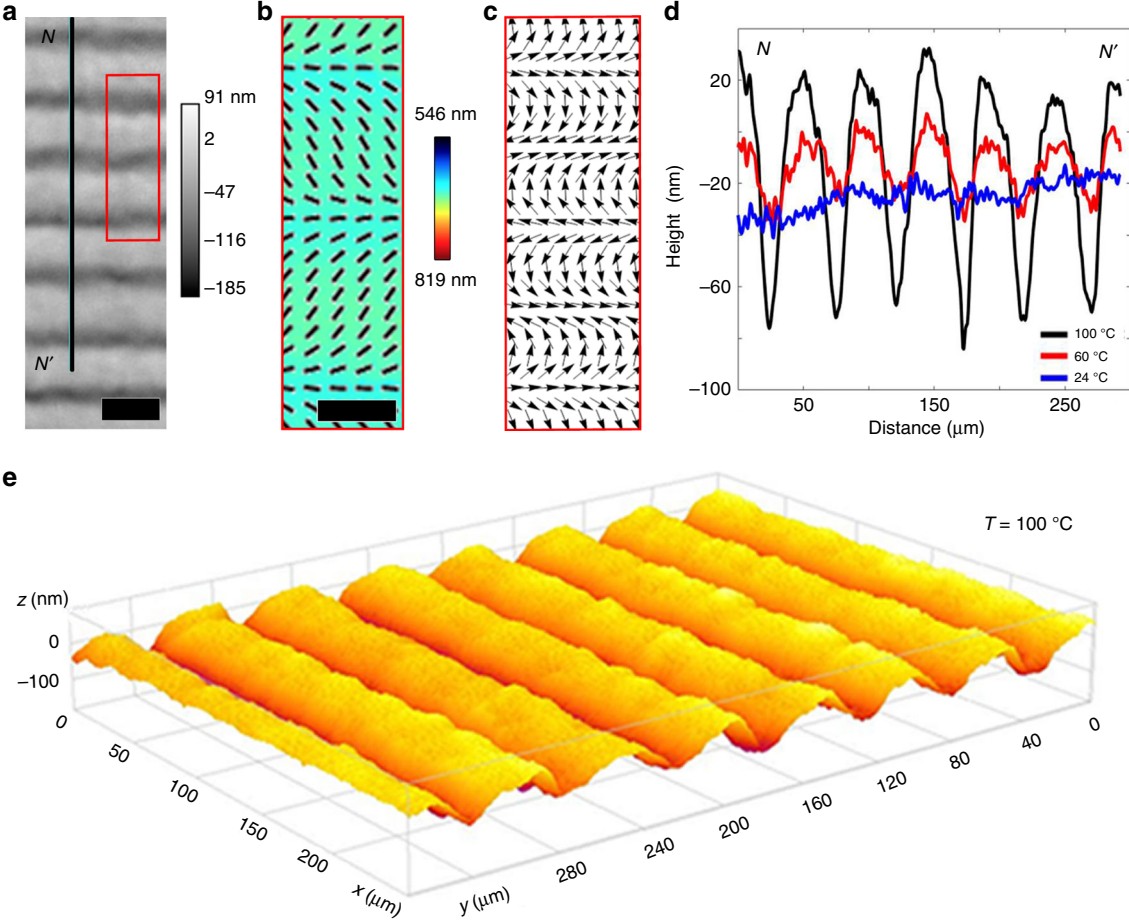

**Fig. 4** Depressions/elevations caused by splay/bend stripes upon heating. **a** DHM image at 100 °C used to extract the surface profile along line *NN′*. Scale bar, 50 μm. **b** PolScope image of a flat LCE coating at 23 °C which maps the optical retardance and the director orientation. Scale bar, 25 μm. **c** Activation force density map calculated for region in (**a**, **b**) outlined by a red box. **d** Surface profiles along line *NN′* in (**a**) at 24, 60, and 100 °C. **e** 3D image of surface topography of LCE at 100 °C observed using DHM

the type of director deformations pre-programmed in the plane of the initially flat sample. For example, in Figs. 1, 2, and 4, splay causes depressions, while bend causes elevations. The relationship is more complicated around $m = +1/2$ defects in Fig. 3, since the vector connecting a depression to an elevation is directed from bend to splay region (Fig. 3a, b). The underlying mechanism can be understood by considering the microscopic response of the LCE to a changing temperature.

Orientational order is coupled to mechanical deformations of LCEs, because of the cross-linking of the polymer network. This coupling results in an anisotropic structure of the network characterized by the so-called step length tensor[1] $l_{ij} = l_\perp \delta_{ij} + (l_\parallel - l_\perp) n_i n_j$. The step length $l$ characterizing the polymer segments connecting cross-linking points is different when measured along $\hat{\mathbf{n}}$ ($l_\parallel$) and perpendicularly ($l_\perp$) to $\hat{\mathbf{n}}$. For $l_\parallel > l_\perp$, the spatial distribution of the step lengths can be represented by a prolate ellipsoid elongated along $\hat{\mathbf{n}}$ (Fig. 5a). If the temperature is raised and the orientational order weakens, the distribution becomes more spherical, i.e., the ellipsoid shrinks along $\hat{\mathbf{n}}$ and expands in two perpendicular directions. Once the nematic order is melted, the ellipsoid becomes a sphere, $l_\parallel^{iso} = l_\perp^{iso} = \bar{l}$ (Fig. 5b).

Morphing of the step length ellipsoid caused by the temperature can be modelled by a force dipole, with two point forces of equal amplitude $F$ directed from the poles of the ellipsoid towards its center (Fig. 5a). For an LCE of a constant volume and small anisotropy, $l_\perp - l_\parallel \ll l_\perp, l_\parallel$, this amplitude can be estimated as

$F \sim \mu \bar{l}(l_\perp - l_\parallel)$, where $\mu$ is the shear modulus of the LCE, on the order of $(10^4-10^5)$ J m$^{-3}$ ref.[21]. Whenever the director field of the LCE changes in space, so do the local axes of the ellipsoids (Fig. 5c, d). The spatial gradients of the step-length tensor produce a vector quantity with the components $f_i = \alpha \partial_j n_i n_j$, which can also be written in the equivalent invariant form as

$$\mathbf{f} = \alpha(\hat{\mathbf{n}} \text{div} \hat{\mathbf{n}} - \hat{\mathbf{n}} \times \text{curl} \hat{\mathbf{n}}) \qquad (2)$$

where $\alpha \sim F/\bar{l}^2 \sim \mu(l_\perp - l_\parallel)/\bar{l}$ is introduced as an activation parameter that describes the local elastic response to the changing temperature; for the sake of simplicity, the estimated value of $\alpha$ corresponds to the complete melting of orientational order. Note that $\alpha$ depends on the dimensionless anisotropy $(l_\perp - l_\parallel)/\bar{l}$ rather than on the absolute values of the step lengths, which stresses a universal character of the elastic response of LCEs with little dependence on the concrete microscopic details[22]. In the order of magnitude, with $(l_\perp - l_\parallel)/\bar{l} \sim 0.1$, one expects $|\alpha| \sim (10^3-10^4)$ J m$^{-3}$. When the temperature of an LCE with $l_\parallel > l_\perp$ increases and the long axes of the polymer ellipsoids shrink, then $\alpha > 0$; in the case of cooling, $\alpha < 0$.

With $\alpha$ defined as above, the vector $\mathbf{f}$ represents a spatially varying activation force density that controls the elastic response of an LCE with a non-uniform director $\hat{\mathbf{n}}(\mathbf{r}) \neq$ const to the external factors such as heating. The occurrence of the force $\mathbf{f}$ is illustrated in Fig. 5c–f for the cases of pure splay and pure bend. For example, in the case of bend, Fig. 5d, f, the point forces of the two

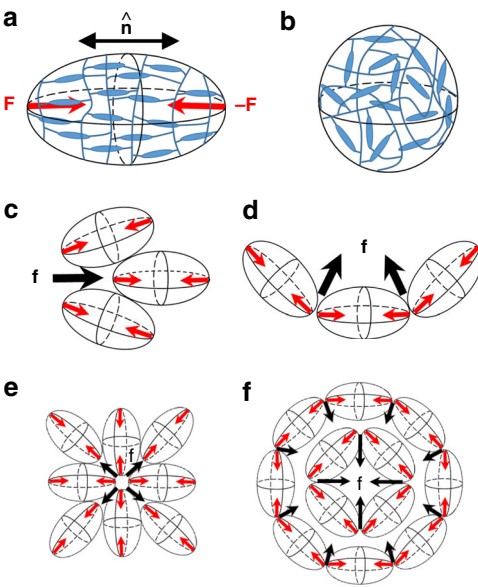

**Fig. 5** Polymer network conformation and occurrence of the activation force. **a** Prolate ellipsoid of polymer network conformations in the nematic phase; the long axis is along the director $\hat{\mathbf{n}}$; during heating, the ellipsoid shrinks along the long axis; in the isotropic phase, it becomes a sphere, as shown in part (**b**); the shrinking ellipsoid is modelled by a pair of forces **F**. **c** Activation force density **f** produced by contracting ellipsoids in the geometry of splay. **d** Activation force density **f** produced by contracting ellipsoids in the geometry of bend. **e** Map of activation forces in the pattern of a radial splay that push the material away from the center towards the periphery upon heating. **f** Map of activation forces in the pattern of a circular bend that push the material from the periphery towards the center

neighboring shrinking ellipsoids that are tilted with respect to each other, produce a net force density **f** along the radius of curvature of the director $\hat{\mathbf{n}}(\mathbf{r})$.

The activation coefficient $\alpha$ and the vector $f_i = \alpha \partial_j n_i n_j$ are similar to the activity coefficient and active current introduced by Aditi Simha and Ramaswamy[23] in the description of flowing active matter with elementary swimming units representing force dipoles of either puller or pusher type; the former are similar to the shrinking LCE ellipsoids (Fig. 5a). Furthermore, the factor $\hat{\mathbf{n}}\,\mathrm{div}\,\hat{\mathbf{n}} - \hat{\mathbf{n}} \times \mathrm{curl}\,\hat{\mathbf{n}}$ is identical to the gradient expression of the flexoelectric polarization in a spatially-nonuniform liquid crystal in which the molecular structure is of quadrupolar symmetry[24]. It is also the same as the gradient part of the active force considered by Green et al.[25] for distorted incompressible active nematic fluids. All these similarities bring under one umbrella rather different phenomena, flow of active fluids, flexoelectric polarization of a distorted nematic liquid crystal and stimuli-responsive LCEs. The connection is not surprising as in all cases the symmetry of elementary units (ellipsoids of shrinking-expanding polymer networks in LCEs) is quadrupolar, and the orientation of these units varies in space. Note that the force **f** in Eq. (2) is not uniquely associated with the heating/cooling and can be used to describe the effect of other stimuli, such as light irradiation, humidity change, etc.

The spatial distribution of the activation force **f** acting in pre-programmed director patterns (Figs. 1e, 2e, 3g, h, and 5e, f) qualitatively explains the deterministic relationships splay → depression and bend → elevation. Consider first the $m = +1$ defect with a radial splay (Figs. 1a and 5e). The force **f** is directed away from the center upon heating, Figs. 1e and 5e. This force transports the matter within the coating. Since the bottom surface of

the coating is affixed to the substrate, the transport results in thinning of the coating. The coating thus should develop a depression, as observed, Fig. 1b–d.

The situation is opposite for the circular bend defect of the same topological charge $m = +1$, Figs. 2a, and 5f. Upon heating, the force **f** moves the material towards the center, Fig. 5f, making the coating thicker in this region, Fig. 2b–d. It is instructive to compare this behavior to a response of an LCE film containing a similar circular bend $m = +1$ in-plane pattern, but having both surfaces free to deform. As demonstrated theoretically by Modes et al.[26], such a free LCE film responds to raising temperatures by bulging out of plane and forming a hollow cone that has an equal probability of protruding upwards or downwards. The top and bottom free surfaces of the film experience similar conical deformations, as clearly seen in the experiments by Ware et al.[7]. In the LCE coating with a fixed substrate, the free surface protrudes only "upwards" and the conical elevation is filled with the material transported from the periphery of the coating by the activation forces **f** in Eq. (2).

In the case of $m = -1$ defects, the force **f** produces four ridges and four valleys, by converging in the bend regions and diverging in the splay regions. For example, the activation force density map in Fig. 1e predicts valleys along the horizontal and vertical directions, and the ridges that are at 45 degrees to the valleys, which is what is seen in the experiment (Fig. 1b, c). Similar profiles are formed around $m = -1$ defects in Fig. 2b, c.

The activation force **f** also helps to understand why the cores of $m = +1/2$ upon heating move towards the splay region: the angular distribution of the force around the $m = +1/2$ defect core is not symmetric, breaking the fore-aft symmetry with a nonzero net force directed towards the splay (tail) region. This effect is yet another demonstration of a deep analogy between the responsive LCEs and active matter such as arrays of vibrating rods[27], living cell cultures[28,29], bacterial colonies[30] and microtubules powered by kinesin motors[31]. In all these systems, $m = +1/2$, unlike their $m = -1/2$, $\pm 1$ counterparts, show a net propulsion in out-of-equilibrium dynamics, either in the direction of bend or splay, depending on whether the active units are extensile or contractile; these types differ in the sign of the activity coefficient $\alpha$. The heated LCE corresponds to a contractile case, $\alpha > 0$; an extensile version with $\alpha < 0$ could be manufactured by polymerizing the nematic LCE at elevated temperatures and then cooling it down. The map of the activation force for $m = +1/2$ suggests that the material is pulled from the bend region towards the splay region (Fig. 3h, j). An exact analytical description of the resulting profile of the LCE coating is not easy to construct, since the solution should account for mass conservation, dynamic coupling of the director field, and rubber elasticity to the material transport, different boundary conditions at the two interfaces, etc.; however, the concept of activation force provides the key insights into the deterministic relationship between the in-plane molecular order and heat-triggered surface profiles.

Note here an interesting similarity in the behavior of LCE coatings and interfacial tissues, or "skins", in living organisms. Very recent research breakthroughs[28,32] demonstrated two key factors controlling cell dynamics in biological tissues such as epithelium, namely, orientational order of cells and presence of topological defects in this orientational order. Activity of individual cells combined with the spatially-varying order leads to compressive-dilative stresses in the tissue that facilitate in-plane and out-of-plane displacements, an effect similar to the observed behavior of LCE coatings.

The described variations of surface profile of LCE coatings are rooted in temperature-induced changes in spatially-varying step-length tensor. At this most general level, the situation is similar to the 3D behavior of LCE thin films with two free surfaces that

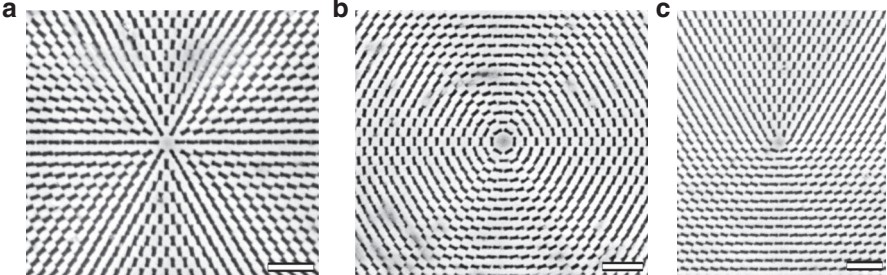

**Fig. 6** Plasmonic metamasks. Scanning electron microscopy images of fragments of metamasks with (**a**) radial defect, $m = +1$ and $\phi_0 = 0$. Scale bar, 1 μm. **b** circular defect, $m = +1$ and $\phi_0 = \pi/2$. Scale bar, 1 μm. **c** $m = +1/2$ defect. Scale bar, 1 μm

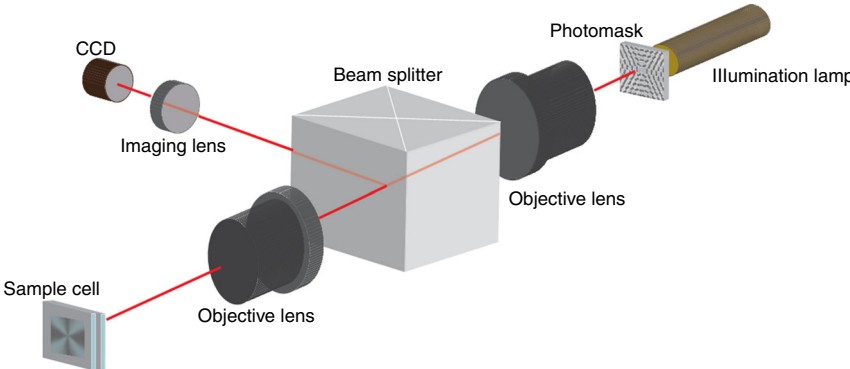

**Fig. 7** Optical set-up for photopatterning. Experimental set-up for photopatterning the director field onto the empty cell consisting of two glass substrates coated with photosensitive layer

develop regions of varying Gaussian and mean curvature, as demonstrated theoretically[12,26,33–38] and experimentally[5–11,39]. There are important differences, however. The theoretical modelling of the LCE free films considers the limit of vanishing thickness; thickness variations are not involved in the mechanism of bending, although they are important in shaping the free films in the regions of maximum curvatures, as described by Modes et al.[26]. The top and bottom surfaces of the free LCE films experience the same deformation, except near the regions where the curvature radii are comparable to the film thickness. Top-down symmetry of the free films produces an uncertainty in the direction of the film's bulging[12,26,33–38]. For example, a free film with a circular bend pattern, $m = +1$, changes its shape upon heating into a hollow cone of a positive Gaussian curvature that can protrude either upwards or downwards with respect to the initial flat plane[26,33,37,38]. The LCE coatings are different, since only one of the surfaces is free. Upon heating, the activation forces produce only an upward elevation in a circular bend pattern of $m = +1$ (Figs. 2b and 5f) and never a downward depression, while a radial splay with $m = +1$ yields a depression (Figs. 1b and 5e) and never an elevation. Furthermore, the activation forces around $m = +1/2$ defects result in an in-plane mobility of these defects in LCE coatings, while in the case of LCE free films, such a mobility has not been described.

To conclude, we demonstrated that the dynamic surface profile of the LCE coatings activated by temperature can be pre-programmed deterministically by inscribing in-plane patterns of orientational order into the initially flat elastomer at the stage of preparation. Deformations such as splay, bend, and their combinations cause different response of the LCE coating, triggering topography changes with local elevations, depressions, and in-plane shifts. The mechanism of the effect is explained by introducing an activation force that makes the description of dynamically addressed LCEs similar to that of active matter. The

proposed approach implies that the activated LCEs can serve as a well-controlled experimental model of active matter, which is a very welcome addition to this rapidly developing field. From the practical point of view, the ability to control 3D shape changes through 2D inscribed patterns of orientational order could be employed to produce coatings with dynamic hydrophobicity/hydrophilicity, coatings that could move microparticles in space according to the underlying topography of their surface, etc.

## Methods

**Photoalignment of substrates**. Glass substrates were first cleaned and treated with UV ozone. Subsequently, the photosensitive material, LIA-S, (0.5%) (DIC Corporation) in dimethylformamide was spin-coated on them and successively baked at 90 °C for 30 min. The cell was assembled using two indium tin oxide coated glass substrates treated with LIA-S, and the spacing was controlled with 5 μm silica spacers (Bangs Laboratories, Inc.) which were pre-mixed with NOA UV71 glue. In order to pre-pattern liquid crystal molecular director onto the treated glass substrates, we used plasmonic metamask (PMM) (Fig. 6) made of rectangular nanoaperture arrays ($100 \times 220$ nm) in Al film[19]. Illuminating PMM with EXFO X-Cite series 120 unpolarized light source creates a pattern of linearly polarized light. The resultant polarized light photoaligns the azobenzene chromophores of the photosensitive layer perpendicularly to the polarization direction of the incident light (Fig. 7)[19]. The polymerized films were probed at 546 nm wavelength using PolScope in order to map the director configuration of an LCE in the $xy$-plane using the optical phase retardance measurements. Note that the typical period of the patterned director in our work is ≈100–200 μm, which is much smaller than the period of structures of free films studied by Ware et al.[7]. The smallest pixel in ref.[7] is about $100 \times 100$ μm, while in our approach is it about $4.5 \times 4.5$ μm or better[19].

**Preparation of the liquid crystalline polymeric coatings**. The chemical structures of the reactive monomers RM82 (25 wt%), RM23 (25 wt%), RM105 (49.2 wt%) (Merck) and photoinitiator Irgacure 819 (0.8 wt%) is shown in Fig. 8. The monomers were homogeneously mixed with the photoinitiator in Dichloromethane overnight at 40 °C. The monomeric mixture was injected into the photoaligned cells in an isotropic phase at 80 °C by capillary action. The samples were than cooled down to the nematic phase and kept at 50 °C during the photopolymerization which was performed using 365 nm LED lamp (Thorlabs M365LP1) for 10 min with intensity of 8.8–18.3 m W/cm². The intensity was

**Fig. 8** Materials. The chemical composition of liquid crystal monomers and photoinitiator used to create the responsive LCEs. **a** RM82 (25 wt%). **b** RM23 (25 wt%). **c** RM105 (49.2 wt%). **d** Irgacure 819 (0.8 wt%)

controlled by LED controller (Thorlabs DC4104) and a collimator (Thorlabs SM2F32-A). After the polymerization, the top and bottom substrates were separated in order to observe the topography of the film attached to a glass substrate.

**Surface topography measurements**. Dynamic optical topography measurements of the partially reflecting surfaces of liquid crystalline polymeric coatings attached to a glass substrate were recorded using Reflection DHM (Lyncée Tec) as a function of temperature. Digital holograms were created using 666 nm monochromatic reference beam which interfered with the object beam received from the polymeric coating. The acquired phase images in real time provided us with the quantitative data for 3D reconstruction of the surface profiles, with the vertical resolution of ≈0.3 nm.

**Data availability**. The data that support the findings of this study are available from the corresponding author upon reasonable request.

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

## Acknowledgements

The work was supported by the NSF grant DMR-1507637, the Netherlands Organization for Scientific Research (NWO; TOP PUNT grant 10018944) and the European Research Council (Vibrate ERC, grant 669991). The scanning electron microscopy data were obtained at the Characterization facility at the Liquid Crystal Institute, Kent State University. The authors thank R. Green, D. Liu, S. Paladugu, S. V. Shiyanovskii, J. Toner, and V. Vitelli for fruitful discussions. O.D.L. and G.B. are thankful for the kind hospitality of Eindhoven University of Technology during their visit to Drs. Broer and Schenning laboratories where preparation and DHM characterization of the LCE coatings were performed.

## Author contributions

G.B. and O.D.L. designed the experiments and wrote the manuscript with the input from all co-authors; G.B. performed the experiments and analysed the data; O.D.L. developed the theoretical model; T.T. assisted with the force field calculations and substrate preparation; Y.G. and Q.-H.W. produced the plasmonic metamasks; M.H. assisted with DHM experiments. A.P.H.J.S. and D.J.B. provided the LCE materials, advised on the materials properties and techniques of LCE preparation and participated in discussing and interpreting the results; O.D.L. supervised the project. All authors contributed to the editing of the manuscript.

## Additional information

**Competing interests:** The authors declare no competing financial interests.

