## [Peer Review File(PDF 546 kb) · Nature Communications]

Reviewers' comments:

Reviewer #1 (Remarks to the Author):

The authors report a study on liquid crystal elastomer (LCE) coatings that can dramatically change topography with changes in temperature. They demonstrate the formation of peaks and valleys, depending on the liquid crystal pattern imprinted during synthesis and the temperature of the film. This work builds on recent developments in the field which have made complex director orientations in LCEs and liquid crystal networks possible. While recent work by White and others in the field have focused on elastomers that exhibit 3-D shape changes, this study is specifically focused on LCE films that are remain adhered to a substrate.

The authors explore three different types of radial director patterns: +1 splay defect, +1 bend defect, and +/- 1/2 defects. They show that +1 splay defects produce a large and reversible depression in the film, while +1 bend defects produce a broader protrusion. The +/- 1/2 defects produce a combination of the two along with a change in the spacing between defects. They further show that defect-free splay/bend patterns exhibit a wavy surface pattern.

The authors furthermore present a logical and intuitive analysis of the observed defects connected to the change in the anisotropy of the polymer networks and expected local strains, which are dependent on the local orientations and change in liquid crystal order parameter.

Overall this is a high-quality study of liquid crystal elastomer films that demonstrate a straightforward approach to generate topographically responsive liquid crystal films. The relationship of the work presented to recent studies by White, Modes on deformations in LCE should be clarified - I was confused initially regarding the significance of the present study until I read the work more carefully and went back to the previous studies. There are also details on the film chemistry and synthesis which should be addressd:

1) The authors should clarify how the present study is different from recent studies by White et al. as in Science 347, no. 6225 (February 27, 2015): 982–84. <https://doi.org/10.1126/science.1261019> and Advanced Materials 25, no. 41 (2013): 5880–85. <https://doi.org/10.1002/adma.201301891>. I believe the major difference is that in these prior studies, White et al. studied elastomers which curved and bent with changes in temperature, while in the current study a coating is formulated which can form potrusions or depressions on the surface. If this is the case, the authors should emphasize this point.

2) The authors present an intuitive and logical explanation for film deformation. The authors should clarify how the model presented is related to a model developed by Modes and Warner in Phys. Rev. E 84, 021711, 2011.

3) The preparation involves polymerization between two glass plates followed by separation of the plates. Is there any treatment applied to one side to ensure the film adheres only to one side? How do the authors achieve a smooth film adhered to only one plate?

4) The chemistry appears to involve crosslinking of the film, but the authors comment that orientation is lost if the sample is heated to 150C - is this truly a liquid crystal elastomer, or is the film not fully crosslinked? Is the Tg for the sample known?

5) The model presented by the authors only relies on the network anisotropy and liquid crystal orientation. Is there a need to account for film adhesion to a substrate? I would guess that this would work to suppress topographical changes in the sample, and film thickness may also have an effect on topography.

6) Some figures are difficult to read. For example, the text in Figures 1b, 1d, 2b, 2d, 3c, 3e, 3f, 4d, and 4e is too small to read clearly.

Overall this is a high quality study that I recommend accepting for publication once the points above have been addressed.

Reviewer #2 (Remarks to the Author):

This paper shows that the dynamic surface profile of the LCE coatings activated by temperature can be pre-programmed deterministically by inscribing in-plane patterns of orientational order into the initially flat elastomer at the stage of preparation. The authors previously developed a plasmonic photomask to record high-resolution 2D linear polarization pattern combined with a thin axis-selective photoresponsive alignment layer. The plasmonic photomask enables to align liquid crystal monomers in an arbitrary pattern. Thermal expansion and contraction in the defects cause topography changes with local elevations and depressions due to molecular alignment pattern in the

defect. Such a phenomenon is quite interesting to broad readers in addition to liquid crystal researchers. Logic flow is appropriate, evident by necessary experimental results. The most important concern for fair assessment of this paper is how superior this paper is to the previous work by White published in Science in 2015. Consideration of the following comments might make the fundamental value of the paper clear.

What is the smallest size of the molecular alignment pattern. In the Science paper a single radial molecular orientation size was 5 mm x 5 mm. With this film reversible actuation is demonstrated.

In the present paper, dynamic surface motion is discussed. Is this phenomenon due to a glass substrate to fix the elastomer? Then surface elevation and depression seem film-thickness dependent. If a free-standing film is prepared and heated, does the film show origami-like actuation?

Does the film thickness change by heating? Heating of in-plane aligned elastomer film must induce in-plane shrinkage. Surface level seems set at 0 before and after heating for comparison in Fig. 1d.

Is the surface elevation in the defect reversible for many cycles? Aspect ratio of the thermo-induced structures needs to be shown.

Explanation of Fig. 5d is misleading. The authors claim that summation of contractions of domains produces elevation in the center. However, contraction of ellipsoids produces perpendicular expansion, which must be a direct power of elevation in the center.

Reviewer #3 (Remarks to the Author):

In this manuscript, the authors precisely controlled the orientation of LC mesogens in LCE matrix by the plasmonic photoalignment technique. The authors then established the relationship between deformation of LCE coatings and LC director field through both experiments and theoretical explanations. The paper was well-organized and experimental data were properly illustrated. A few minor changes will be needed to improve the manuscript as noted below.

1) The authors started the manuscript with discussion of biological relevance of their LCE coatings. However, the entire manuscript has nothing to with biological systems.

2) The authors stated the challenge of programmable control of topographic deformation of LCE coatings in line 59. However, deterministic pre-programming has been presented by several recent literature on this topic, although authors did not cite them, including Xia Y. et al. *Adv. Mater.*, 2016, 28 (43), 9637; Ware, T. H. et al. *ACS Macro Lett.*, 2015, 4 (9), 942; *ACS Appl. Mater. Interfaces*, 2017, 9 (42), 37332.

It is suggested that the authors focus on discussion what's the difference between programmable deformation of LCE films vs. LCE coatings.

3) The authors showed the correlation of splay-depression and bend-elevation throughout the paper. It is not clear how the substrate will affect the depression and elevation force. Then what's the minimum thickness of LCE coating by such method?

4) In lines 103-105, it is stated that "The dynamics of surface topography is completely reversible when the temperature is cycled between the room temperature and the maximum temperature 104 of around 120°C. The reversibility is lost if the material is heated into an isotropic state with no orientation order (at about 150°C)."

So why 120°C, which is below the TNI (150 °C)?

why was the reversibility lost when heating above TNI? Since the LCE coatings were polymer network, the surface relief coating should be reversible upon cooling.

5) In lines 101 and 102, the authors addressed "m = -1 defects" and cited Figure 1b,c and Figure 2b,c. However, figure captions in Figure 1 and 2 both showed "+1 defect".

6) From line 124 to line 126, the authors discussed about both "m = +1/2 and m = -1/2 defects" and cited Figure 3a. It is suggested that the authors specify in Figure 3a which one is "m = +1/2" and which one is "m = -1/2".

7) In line 138, the authors talked about topography along line TT' in Figure 3c. Following the explanation in this paper, topography along line TT' should have more than three splay and bend regions given that line TT' crossed three defect cores including "m = -1/2 defect", then Figure 3c

should show more elevation/depression pairs than what is shown in the paper. Need explanation/clarification here.

8) In line 171, figure caption of Figure 4 stated as “splay director deformations”, but bend deformations should also exist.

9) Some references were not cited properly. For example, in line 46-48 & 51 the authors gave examples of thermal induced shaping of LCE and cited reference 4, 5, 7-11. However, reference 5 was about electrically induced actuation and reference 8 was about photomechanical responses.

Reviewers' comments:

Reviewer #1 (Remarks to the Author):

1.1. Overall this is a high-quality study of liquid crystal elastomer films that demonstrate a straightforward approach to generate topographically responsive liquid crystal films. The relationship of the work presented to recent studies by White, Modes on deformations in LCE should be clarified - I was confused initially regarding the significance of the present study until I read the work more carefully and went back to the previous studies. There are also details on the film chemistry and synthesis which should be addressed:

The authors should clarify how the present study is different from recent studies by White et al. as in *Science* 347, no. 6225 (February 27, 2015): 982–84. <https://doi.org/10.1126/science.1261019> and *Advanced Materials* 25, no. 41 (2013): 5880–85. <https://doi.org/10.1002/adma.201301891>. I believe the major difference is that in these prior studies, White et al. studied elastomers which curved and bent with changes in temperature, while in the current study a coating is formulated which can form protrusions or depressions on the surface. If this is the case, the authors should emphasize this point.

Answer 1.1. Yes, the principal difference is that the manuscript deals with coatings with only one surface being free to deform, while Ware et al (*Science* 347, 982 (2015)), McConney et al (*Adv.Mat.* 25, 5880 (2013)), Ambulo et al (*Acs Appl Mater Inter* 9, 37332 (2017)), Ware et al (*Acs Macro Lett* 4, 942 (2015)), Xia et al (*Adv. Mat.* 28, 9637 (2016)) and Godman et al (*Acs Macro Lett* 6, 1290 (2017)) considered films with two free surfaces. When the free film deforms, the

changes of its thickness are not important for understanding the mechanisms of bending. Both the top and bottom surfaces of a free film experience similar deformations. In our case of the coatings, the thickness changes represent the prime effect. There is also a principal difference in the deterministic relationship between the surface profile and the in-plane director patterns. A free film with circular patterns in above mentioned references has the same probability of popping “up” or popping “down” upon heating. The coatings explored in our manuscript show a much higher degree of deterministic relationship between the in-plane director deformations and the profile changes. Namely, the similar circular patterns, Fig.2, always develop elevations (protrusions) and never depressions. The depressions appear only when the coating contains radial patterns instead of the circular ones. As suggested by Reviewer 1, we added a discussion of differences on pages 12-14 of the revised manuscript.

1.2. The authors present an intuitive and logical explanation for film deformation. The authors should clarify how the model presented is related to a model developed by Modes and Warner in Phys. Rev. E 84, 021711, 2011.

Answer 1.2. The model by Modes and Warner (PRE 84, 021711 (2011)), as well as other theoretical models (Aharoni et al, PRL 113, 257801 (2014), Mostajeran PRE 91, 062405 (2015)) focus on stimuli-induced metric changes of the films with *two free surfaces in the limit of vanishing thickness*. The models predict how these free films deform, but the *variation of film’s thickness* is not the main issue of the mechanism, as both surfaces deform in a similar way. The main point of our model is that heating produces physical forces (that we call activation forces) that move the material within the coating, resulting in *significant thickness changes, such as elevations and depressions*. We are not aware of any work that introduces the activation forces, Eq.(2) in liquid crystal elastomers. We demonstrate that the concept of activation forces explains the observed deterministic relationship between the director deformations in the plane of the coating and the thickening/thinning of the coating. The introduced concept of activation forces makes an important connection between the physics of elastomers and active matter.

The two models predict different results. As already discussed, a circular pattern in Modes and Warner model can cause the free film to pop “up” or “down”; this ambiguity has been confirmed in numerical simulations by Konya et al (*Front. Matter* 3, 24 (2016)). In contrast, in our model, the circular pattern can cause only elevation (thickening or protrusion “up”) of the film on heating. Furthermore, a defect of strength $+1/2$ in the model by Modes and Warner produces a single conical shape popped either up or down from the initially flat film. In our case, a $+1/2$ defect produces an elevation/depression pair that also shows a shift within the coating’s plane. As suggested by the Reviewer, we added a discussion of the main differences in the text of the revised manuscript on pages 12-14, see also our reply 1.5 below.

1.3. The preparation involves polymerization between two glass plates followed by separation of the plates. Is there any treatment applied to one side to ensure the film adheres only to one side? How do the authors achieve a smooth film adhered to only one plate?

Answer 1.3. Preparation of elastomers with a preprogrammed director require us to use very special photosensitive surface aligning agents at both glass plates. To ensure that the director is uniform across the initial flat coating, we must use identical substrates. This at present precludes us from gluing the elastomers to one of the substrates. However, even without such a preferential

treatment of one of the substrates, we succeeded in obtaining sufficiently large areas of uniform coatings by tearing the cell apart, as the adhesion forces are never identically strong at the two substrates.

1.4. The chemistry appears to involve crosslinking of the film, but the authors comment that orientation is lost if the sample is heated to 150C - is this truly a liquid crystal elastomer, or is the film not fully crosslinked? Is the T_g for the sample known?

Answer 1.4. The T_g of our mixture is about 50°C, as stated in the manuscript, page 3. The statement about the “isotropic phase” was erroneous, the phrase “The reversibility is lost if the material is heated into an isotropic state with no orientational order (at about 150°C)” is now replaced with “The most pronounced profile deformations, such as depressions in Fig. 1b,d and elevations in Fig. 2b,d can lose their complete reversibility if the material is heated to temperatures above 150°C, presumably because of the developed irreversible strains, similarly to the case reported by Ware et al for thermally addressed LCE films with two free surfaces (*Science* 347, 982 (2015))”. We verified that the film preserves its orientational order by detecting well-defined birefringent patterns at temperatures as high as 200°C.

1.5. The model presented by the authors only relies on the network anisotropy and liquid crystal orientation. Is there a need to account for film adhesion to a substrate? I would guess that this would work to suppress topographical changes in the sample, and film thickness may also have an effect on topography.

Answer 1.5. Yes, a complete treatment of the problem would require to consider explicitly the different boundary conditions, concrete dependence on the thickness of coatings. As stated in the manuscript, “An exact analytical description of the resulting profile ...should account for ... different boundary conditions at the two interfaces.” We emphasized the role of different boundary conditions by detailing the description on page 12 in the revised manuscript and contrasting the behavior to that of LCE films with two free surfaces.

At the same time, we stress that a complete solution of the temperature dependent surface profile could be achieved only by numerical simulations. There is no simple analytical expression to explain the transport of mass when the activation forces are accompanied by other factors. Because of that, we focus on the activation forces as they allow us, in this Reviewer’s words, “(to) present an intuitive and logical explanation for film deformation.”

1.6. Some figures are difficult to read. For example, the text in Figures 1b, 1d, 2b, 2d, 3c, 3e, 3f, 4d, and 4e is too small to read clearly.

Answer 1.6. Thank you, we have improved the figures as suggested.

1.7. Overall this is a high quality study that I recommend accepting for publication once the points above have been addressed.

Answer 1.7. We thank the Reviewers for the comments that helped us to present the material better in the revised manuscript.

Reviewer #2 (Remarks to the Author):

2.1. This paper shows that the dynamic surface profile of the LCE coatings activated by temperature can be pre-programmed deterministically by inscribing in-plane patterns of orientational order into the initially flat elastomer at the stage of preparation. The authors previously developed a plasmonic photomask to record high-resolution 2D linear polarization pattern combined with a thin axis-selective photoresponsive alignment layer. The plasmonic photomask enables to align liquid crystal monomers in an arbitrary pattern. Thermal expansion and contraction in the defects cause topography changes with local elevations and depressions due to molecular alignment pattern in the defect. Such a phenomenon is quite interesting to broad readers in addition to liquid crystal researchers. Logic flow is appropriate, evident by necessary experimental results.

Answer 2.1. We thank the Reviewer for stressing that the work is interesting to a broad audience.

2.2. The most important concern for fair assessment of this paper is how superior this paper is to the previous work by White published in *Science* in 2015. Consideration of the following comments might make the fundamental value of the paper clear.

Answer 2.2. The paper by White et al (*Science* 347, 982 (2015)) deals with the films of elastomers in which the two surfaces are free. Upon heating, the film with circular director configurations forms hollow conical protrusions that pop either up or down with respect to the initial plane. The thickness changes of the film are not important as both surfaces experience similar deformations.

In our work, we deal with coatings in which one surface is immobilized. Thermal action changes the local thickness of the film, resulting in profiles with elevations and depressions, which is the main result and the main difference with the work of White. There are also other differences, presented in details in our answer 1.1, 1.2, 1.5. We added the relevant discussion of differences in the revised text on pages 12-14.

2.3. What is the smallest size of the molecular alignment pattern. In the *Science* paper a single radial molecular orientation size was 5 mm x 5 mm. With this film reversible actuation is demonstrated.

Answer 2.3. In our patterns, the period of the pattern (or the typical “size” of the single radial or circular defect) is much smaller, $\approx 100\text{-}200\ \mu\text{m}$, Fig.1 and 2. In principle, it can be made larger, i.e. 5 mm, by using appropriate objectives. Furthermore, there is a difference in resolution. According to Ware et al (*Science* 347, 982 (2015)), “226 distinct director orientations are patterned into 21,350 voxels of material, each 100 by 100 μm in area”. The resolution of our molecular alignment pattern technique is much higher, $\sim 4.5\ \mu\text{m}$, as described in Guo et al (*Adv Mater* **28**, 2353-2358 (2016)) and confirmed by optical microscopy studies of the resulting films with the Figures presented in the manuscript. We add the following clarification in the revised Supplementary section on page 1: “Note that the typical period of the patterned director in our work is $\approx 100\text{-}200\ \mu\text{m}$, which is much smaller than the period of structures of free films studied by Ware et al ². The smallest pixel in Ref. 2 is about 100 x 100 μm , while in our approach it is about 4.5 μm x 4.5 μm or better ¹.”

2.4. In the present paper, dynamic surface motion is discussed. Is this phenomenon due to a glass substrate to fix the elastomer? Then surface elevation and depression seem film-thickness dependent. If a free-standing film is prepared and heated, does the film show origami-like actuation?

Answer 2.4. Our surface coatings at present are too thin to make a robust free film. Note that the works with films having two free surfaces are dealing with much thicker (50 μm and more) films. Nevertheless, we understand the importance of expanding our studies to free films and currently developing approaches to make these films more robust.

2.5. Does the film thickness change by heating? Heating of in-plane aligned elastomer film must induce in-plane shrinkage. Surface level seems set at 0 before and after heating for comparison in Fig. 1d.

Answer 2.5. The prime result is that the film develops local elevations and depressions upon heating, which implies shrinkages and expansions in the plane of the film, as illustrated by the maps of activation force in Figs. 1e, 2e, 3g,h, 4c. Conservation of mass might result in some changes of the overall thickness, but since the elevations/depressions are localized and their amplitude is only about 10% of the overall thickness of the film, these modifications of the overall thickness are hard to detect at the moment.

2.6. Is the surface elevation in the defect reversible for many cycles? Aspect ratio of the thermo-induced structures needs to be shown.

Answer 2.6. The surface elevations and depressions are reversible, if the maximum temperature is limited by about 120°C. We successfully repeated the experiments up to 4-5 cycles. However, if the topographic deformations are too extreme – typically at very high temperatures (i.e. 150°C), the LCE coating may experience very high strain levels which will cause a hysteresis of the surface profile. Per Reviewer's suggestion, we included the aspect ratio of the thermo-induced structures on page 4.

2.7. Explanation of Fig. 5d is misleading. The authors claim that summation of contractions of domains produces elevation in the center. However, contraction of ellipsoids produces perpendicular expansion, which must be a direct power of elevation in the center.

Answer 2.7. We think that we did not make our point clear on this issue in the original manuscript. We added parts 'e' and 'f' to Fig. 5 and updated the caption to explain things better in the revised version. As seen in Fig. 5f, contraction of ellipsoids produces a radial inward force that leads to an elevation.

Reviewer #3 (Remarks to the Author):

3.1. In this manuscript, the authors precisely controlled the orientation of LC mesogens in LCE matrix by the plasmonic photoalignment technique. The authors then established the relationship between deformation of LCE coatings and LC director field through both experiments and

theoretical explanations. The paper was well-organized and experimental data were properly illustrated. A few minor changes will be needed to improve the manuscript as noted below.

Answer 3.1. We thank the Reviewer for the positive feedback regarding the organization of the manuscript and the presentation of our data.

3.2. The authors started the manuscript with discussion of biological relevance of their LCE coatings. However, the entire manuscript has nothing to do with biological systems.

Answer 3.2. Recent discoveries in the behavior of biological coatings such as epithelium (Saw et al, *Nature* 544, 212-216 (2017) and Kawaguchi et al, *Nature* 545, 327-331 (2017)) demonstrated changes of surface topography and in-plane material transport. We observed similar physical effects in our artificial elastomers and decided to mention the connection. The connection is substantiated not only by the observed topography changes but also by the underlying physics. The activation force we introduce is very similar to the active force considered in numerous “active” systems many of which are of biological nature. We cited many analogies between the elastomers and active biological matter (cell cultures, bacterial colonies, microtubules) in the discussion on page 13. We thus would like to keep the mentioning of the biological coatings in the Introduction since it can also inspire other researchers. We modified the last paragraph of the Introduction to make the connection clearer.

3.3. The authors stated the challenge of programmable control of topographic deformation of LCE coatings in line 59. However, deterministic pre-programming has been presented by several recent literature on this topic, although authors did not cite them, including Xia Y. et al. *Adv. Mater.*, 2016, 28 (43), 9637; Ware, T. H. et al. *ACS Macro Lett.*, 2015, 4 (9), 942; *ACS Appl. Mater. Interfaces*, 2017, 9 (42), 37332.

Answer 3.3. We thank the reviewer for pointing out these important works and added all of them to the list of references. As suggested by the Reviewer, we added a discussion of the main differences between LCE free standing films and the LCE coatings in the text of the revised manuscript on pages 12-14.

3.4. It is suggested that the authors focus on discussion what's the difference between programmable deformation of LCE films vs. LCE coatings.

Answer 3.4. We are thankful for this suggestion and added a corresponding discussion to the text, on pages 12-14.

3.5. The authors showed the correlation of splay-depression and bend-elevation throughout the paper. It is not clear how the substrate will affect the depression and elevation force. Then what's the minimum thickness of LCE coating by such method?

Answer 3.5. We agree that the nature of the substrate would influence the forces. However, at this stage we cannot work with different substrates since the method of plasmonic photoalignment requires us to use transparent optically isotropic substrates such as glass (plastics would not work since they are birefringent) and the surface of these substrates needs to be covered with a layer of

a photosensitive dye. The observed amplitude of surface topography changes is about 10% of the coating's thickness, as follows from our observations for coatings with the thickness in the range 5-50 μm . For these thicknesses, the profile changes are readily observable by optical, holographic and atomic force microscopy. By making coatings much thinner, say, at a submicron level, it would be difficult to accurately measure the range of profile changes because of the limits of resolution and weaker signal/noise ratio. Nevertheless, we plan the studies for the future. At the moment, our answer is that the coatings of a few micrometers still show the effects we observe and describe in details for 5 μm .

3.6. In lines 103-105, it is stated that "The dynamics of surface topography is completely reversible when the temperature is cycled between the room temperature and the maximum temperature 104 of around 120°C. The reversibility is lost if the material is heated into an isotropic state with no orientation order (at about 150°C)." So why 120°C, which is below the TNI (150 °C)? why was the reversibility lost when heating above TNI? Since the LCE coatings were polymer network, the surface relief coating should be reversible upon cooling.

Answer 3.6. The statement about the "isotropic phase" was erroneous, the phrase "The reversibility is lost if the material is heated into an isotropic state with no orientational order (at about 150°C)" should be replaced with "The most pronounced profile deformations, such as depressions in Fig. 1b,d and elevations in Fig. 2b,d can lose their complete reversibility if the material is heated to temperatures above 150°C, presumably because of the developed irreversible strains, similarly to the case reported by Ware et al for thermally addressed LCE films with two free surfaces (*Science* 347, 982 (2015))". We verified that the film preserves its orientational order by detecting well-defined birefringent patterns at temperatures as high as 200°C.

3.7. In lines 101 and 102, the authors addressed "m = -1 defects" and cited Figure 1b,c and Figure 2b,c. However, figure captions in Figure 1 and 2 both showed "+1 defect".

Answer 3.7. Thank you, we modified the captions to stress the presence of both +1 and -1 defects.

3.8. From line 124 to line 126, the authors discussed about both "m = +1/2 and m = -1/2 defects" and cited Figure 3a. It is suggested that the authors specify in Figure 3a which one is "m = +1/2" and which one is "m = -1/2".

Answer 3.8. As suggested by the Reviewer, Figure 3a has been modified, and the appropriate labels for m = +1/2 and m = -1/2 defects have been inserted.

3.9. In line 138, the authors talked about topography along line TT' in Figure 3c. Following the explanation in this paper, topography along line TT' should have more than three splay and bend regions given that line TT' crossed three defect cores including "m = -1/2 defect", then Figure 3c should show more elevation/depression pairs than what is shown in the paper. Need explanation/clarification here.

Answer 3.9. Figure 3b illustrates line TT' that passes through three defects cores: -1/2, +1/2 and -1/2. It is correct that each -1/2 defect has three splay/bend regions if one circumnavigate the core along an azimuthal direction (and Fig. 3b,d clearly show three elevations and three depressions

around each $-1/2$ defect). However, the line TT' crosses only one splay and only bend region when it passes through a $-1/2$ defect. As a result, there are only three elevations and three depressions along the shown extension of the TT' line. This point is clearly seen in Fig. 3b, in which the line TT' goes through three bright regions (elevations) and three dark regions (depressions).

3.10. In line 171, figure caption of Figure 4 stated as “splay director deformations”, but bend deformations should also exist.

Answer 3.10. Yes, we agree and thank for noticing the error, it was an oversight in writing a figure 4 caption, we corrected it to “Surface topography deformations produced by splay-bend director deformations.”

3.11. Some references were not cited properly. For example, in line 46-48 & 51 the authors gave examples of thermal induced shaping of LCE and cited reference 4, 5, 7-11. However, reference 5 was about electrically induced actuation and reference 8 was about photomechanical responses.

Answer 3.11. We thank the Reviewer for noticing a discrepancy in the references. We corrected the list of references.

Summary of Changes

1) In the Introduction, we modified the last paragraph to address the question **3.2.** by Reviewer 3:

“In this work, we demonstrate how to program similar displacements and dynamics of surface profile in an artificial skin, made of a liquid crystal elastomer (LCE). The program is written as a pattern of molecular orientation in the plane of the LCE coating that is initially flat. When activated by temperature, the coating changes its surface profile as prescribed by the in-plane pattern, by moving the material within and out-of-plane. The displacements are deterministically related to the molecular orientation pattern. For example, circular bend of molecular orientation causes elevations, while radial splay causes depressions of the coating.”

2) To address issues raised by Reviewers 1 and 3, we added the following text on page 4:

“The most pronounced profile deformations, such as depressions in Fig. 1b,d and elevations in Fig. 2b,d can lose their complete reversibility if the material is heated to temperatures above 150°C , presumably because of the developed irreversible strains, similarly to the case reported by Ware et al for thermally addressed LCE films with two free surfaces⁹.”

3) To address the question **2.6.** raised by Reviewer 2 about the aspect ratio, we added the following on page 4:

“The typical aspect ratio for depressions in Fig. 1b,d and elevations in Fig. 2b,d, determined at as the depth/height of the deformation divided by its full width at half-depth/height, was about $200\text{ nm}/10$, or 0.02.”

4) To address the question 2.7. we added new parts ‘e’ and ‘f’ to Fig.5 and also added the following to the caption: “e, Map of activation forces in the pattern of a radial splay that push the material away from the center towards the periphery upon heating. f, Map of activation forces in the pattern of a circular bend that push the material from the periphery towards the center.”

5) To address the issue raised Reviewer 3, we updated the captions in Figs 1 and 2, by stating “produced upon heating by a lattice of alternating +1 and -1 defects”

6) To address the issue raised by Reviewer 3, we updated the caption of Fig.4, by stating “upon heating by splay-bend”

7) To address the issue of comparison between our results for coatings and the behavior of films with two free surfaces, we added several paragraphs in the discussion section.

Page 12:

“Consider first the $m = +1$ defect of a radial splay type (Fig. 1a, 5e). The force \mathbf{f} associated with this deformation is directed away from the center upon heating, Fig. 1e and Fig. 5e. This force transports the matter within the coating. Since the bottom surface of the coating is affixed to the glass substrate, the transport results in lateral removal of the material and thinning of the coating. The coating thus should develop a depression around the radial $m = +1$ defect, as observed experimentally, Fig. 1b,c,d.

The situation is opposite for the circular bend defect of the same topological charge $m = +1$, Fig. 2a, 5f. Upon heating, the force \mathbf{f} moves the material towards the center, Fig. 5f, making the coating thicker in this region, as observed experimentally, Fig. 2 b,c,d. It is instructive to compare this behavior to a response of an LCE film containing a similar circular bend $m = +1$ in-plane pattern, but having both surfaces free to deform. As demonstrated theoretically by Modes, Bhattacharya and Warner²⁸, such a free LCE film responds to raising temperatures by bulging out of plane and forming a hollow cone that has an equal probability of protruding upwards or downwards. The top and bottom free surfaces of the film experience similar conical deformations, as clearly seen in the experiments by Ware et al⁹. In the LCE coating with a fixed substrate, the free surface protrudes only “upwards” and the conical elevation is filled with the material transported from the periphery of the coating by the activation forces \mathbf{f} in Eq.(2).”

Pages 13-14:

“however, the concept of activation force provides the key insights into the deterministic relationship between the in-plane molecular order and heat-triggered surface profiles.

The described variations of surface profile of LCE coatings are rooted in temperature-induced changes in spatially-varying step-length tensor. At this most general level, the situation is similar to the 3D behavior of LCE thin films with two free surfaces that develop regions of varying Gaussian and mean curvature, as demonstrated theoretically^{14,28,33-38} and experimentally^{7-13,39}. There are important differences, however. The theoretical modelling of the LCE free films considers the limit of vanishing thickness; thickness variations are not involved in the mechanism of bending, although they are important in shaping the free films in the regions of maximum curvatures, as described by Modes, Bhattacharya and Warner²⁸. The top and bottom surfaces of the free LCE films experience the same deformation, except near the regions where the curvature radii are comparable to the film thickness. Top-down symmetry of the free films produces an uncertainty in the direction of the film's bulging^{14,28,33-38}. For example, a free film with a circular bend pattern, $m = +1$, changes its shape upon heating into a hollow cone of a positive Gaussian curvature that can protrude either upwards or downwards with respect to the initial flat plane^{28,33,37,38}. The LCE coatings are different, since only one of the surfaces is free. The activation forces produce only an upward elevation in a circular bend pattern of $m = +1$ (Fig. 2b, 5f) and never a downward depression, while a radial splay with $m = +1$ yields a depression (Fig. 1b, 5e) and never an elevation. Furthermore, the activation forces around $m = +1/2$ defects result in an in-plane mobility of these defects in LCE coatings, while in the case of LCE free films, such a mobility has not been described.”

8) We also added references 10-13, 28, 33-39 as requested.

We hope that the revised manuscript addresses properly all the concerns and can be accepted for publication.

REVIEWERS' COMMENTS:

Reviewer #1 (Remarks to the Author):

The authors have addressed all questions raised during my previous review and modified the manuscript accordingly. I recommend acceptance for publication.

Reviewer #2 (Remarks to the Author):

I read the revised manuscript, and it seems that the answers and revision in the manuscript is satisfactory. Out of curiosity, what a change in Young's modulus is induced in the surface after a topological change?

Reviewer #3 (Remarks to the Author):

The authors properly addressed all referees' concerns in their revision. No further changes are needed. It is suggested to be published as it is.

Reviewers' comments:

Reviewer #1 (Remarks to the Author): The authors have addressed all questions raised during my previous review and modified the manuscript accordingly. I recommend acceptance for publication.

Response. We thank the Reviewer for the recommendation of our manuscript for publication.

Reviewer #2 (Remarks to the Author): I read the revised manuscript, and it seems that the answers and revision in the manuscript is satisfactory. Out of curiosity, what a change in Young's modulus is induced in the surface after a topological change?

Answer. We thanks the Reviewer for accepting our revisions. Unfortunately, we cannot provide an answer to this question regarding Young's modulus as of today. However, we plan new experiments that might potentially provide an answer.

Reviewer #3 (Remarks to the Author): The authors properly addressed all referees' concerns in their revision. No further changes are needed. It is suggested to be published as it is.

Response. We thank the Reviewer for suggesting to publish our manuscript.